# Psychological Inflexibility in People with Chronic Psychosis: The Mediating Role of Self-Stigma and Social Functioning

**DOI:** 10.3390/ijerph182312376

**Published:** 2021-11-25

**Authors:** Ana González-Menéndez, Tatiana Arboleya Faedo, David González-Pando, Nuria Ordoñez-Camblor, Elena García-Vega, Mercedes Paino

**Affiliations:** 1Department of Psychology, University of Oviedo, 33003 Oviedo, Spain; elenagv@uniovi.es (E.G.-V.); mpaino@uniovi.es (M.P.); 2ISPA-Health Research Institute of the Principality of Asturias, 33011 Oviedo, Spain; Tatiana.arboleya@sespa.es (T.A.F.); gonzalezpdavid@uniovi.es (D.G.-P.); 3Faculty of Health Sciences, University of Burgos, 09001 Burgos, Spain; nordonez@ubu.es

**Keywords:** psychological inflexibility, self-stigma, social functioning, severity of psychosis

## Abstract

Psychosis is associated with self-stigmatization and loss of social functioning that increase the severity of the disorder. Psychological inflexibility (PI)—an individual’s tendency to suppress undesirable private events—plays a fundamental role in the emergence and worst prognosis of psychosis. The main objective of this study was to analyze whether self-stigma and social functioning mediate the association of PI with the severity of psychosis in adults with chronic schizophrenia. The study was carried out with a sample of 103 outpatients. The Acceptance and Action Questionnaire, the Internalized Stigma of Mental Illness Scale, and the Social Functioning Scale were used for clinical assessments. Data analyses were performed by using the PROCESS macro for SPSS. Results showed that the link between PI and the severity of psychosis is not direct, but is better explained by mediation of the self-stigma and social functioning of those assessed. PI also predicts worse social functioning without the need to take self-stigma into account. Moreover, self-stigma alone does not predict the severity of psychotic symptoms; this relationship has to be mediated by social functioning. These findings suggest that interventions designed to increase psychological flexibility, such as Acceptance and Commitment Therapy (ACT), may offer an alternative to attenuate the negative impact of self-stigma and to improve the social functioning.

## 1. Introduction

Mental illness stigma, which includes negative attitudes, beliefs, and actions directed towards persons with severe mental illness, has been implicated in a host of negative consequences. Specifically, stigmatization of people with a diagnosis of schizophrenia is a documented process of interest to research because of its repercussions on various domains of social functioning, quality of life, and social support received and perceived by those diagnosed [1,2,3]. The stigma also affects symptom management, understanding and awareness (insight) of the disorder, and the commencement of and adherence to available treatments [4]. It is usually described as prejudice and discrimination caused by negative stereotypes of those who are associated with the diagnostic label (e.g., people who are dangerous, unpredictable, incompetent) [5,6]. Occasionally, those who are identified as part of a stigmatized group endorse such stereotypes, anticipating social rejection and incorporating such viewpoints into their own systems of values [7] (i.e., self-stigma or internalized stigma). Self-stigma is a prominent feature of people living with psychotic symptoms (i.e., people with psychosis) and their families [8], and can become a barrier to recovery and the pursuit of personally valuable and important goals [9]. 

Although it is not yet clear what the relationships between self-stigma and adverse results observed in different measures of recovery and functioning are [10], the reduction in self-stigma has been raised as a potentially important treatment objective. Within the scope of treatment for psychosis, various stigma-reduction programs have been developed, mostly within a psychoeducational framework that is often based on cognitive behavioral principles. Some interventions targeting self-stigma attempt to modify the content of stigmatizing thoughts in order to change behavior [11]. Other studies have provided promising findings, such as significant improvement in engulfment, hopelessness, quality of life, self-esteem, and personal recovery [12,13,14]. 

However, a number of studies have reported no impact on their primary outcome measures of internalized stigma [14,15,16,17]. Seidman et al. [18], for example, observed positive results in psychological health (e.g., behavior-seeking help), with no changes in self-stigma, which suggests that some form of psychological rigidity may be affecting the relationship between self-stigmatizing thoughts and adaptive behavior; that is, undesirable thoughts, beliefs, emotions, and other internal events are not always enough to influence or dictate behavior. 

This conceptualization is consistent with the psychological inflexibility model proposed by Hayes et al. [19] to explain the development and maintenance of psychological problems. Psychological inflexibility (PI) refers to an individual’s tendency to suppress or change the form and frequency of undesirable private events, such as emotions, thoughts, behaviors, or bodily sensations, in order to cope with and regulate rising negative emotions [20]. PI was introduced within the context of Acceptance and Commitment Therapy (ACT) [21], a contextual cognitive behavioral therapy that emphasizes a non-defensive relationship with symptoms and encourages the pursuit of a life concentrated on personal values. 

The opposite of PI is the acceptance of private experiences and psychological flexibility. Psychological flexibility allows individuals to experience private events in a way that is accepting, defused, and mindful, and allows individuals to act on personal goals regardless of private events [22]. Reduction in hospitalization rates and distress, as well as improved coping without eliminating or reducing symptoms, have been observed following ACT treatment in psychosis patients [23,24,25]. 

Around 1/3 of patients with psychosis achieve clinical remission, but functional recovery may lag behind, often leading patients to significant deficits in social functioning [26] (e.g., lack of socially useful activities and difficulties in personal and social relationships). The way patients respond to symptoms may moderates the relationship between emotional characteristics and functional recovery measures [27]. Identifying risk factors for poor social functioning may help identify those patients who are most at risk of entering the spiral of severe psychosis. In addition, whether the risk factors that affect social functioning are modifiable or not, suggests treatment for addressing such mechanisms and thereby improve its results [28]. 

In chronic psychosis, PI has been demonstrated to be a predictor of self-stigma [29,30] and negative results in various social-functioning domains [31]. Interventions targeting acceptance and psychological flexibility (such as ACT) have been shown to increase resistance to stigma [32], promoting personal recovery [33] and reducing the pernicious effects of self-stigma in those diagnosed with schizophrenia [34,35]. 

We hypothesized that people with chronic psychosis who were more inflexible would be more likely to experience thoughts (in this case, stigmatizing thoughts) as distressing events that they must escape from. Furthermore, given that stronger self-stigmatization predicts worse social functioning [36], we hypothesized that, as a consequence of its effect on the self-stigma experience, greater psychological inflexibility would predict decreased social functioning and more severe psychosis. 

Our first objective was to study whether self-stigma and social functioning mediate the association between PI and the severity of psychosis in a sample of adults with chronic psychosis. Overall, we expected higher levels of PI to be associated with the severity of psychosis (Figure 1, H1). Nevertheless, our hypothesis proposed that the total effect would be influenced by the effects of psychological inflexibility on the self-stigma experience as well as on social functioning; that is, PI predicts internalized stigma of people with psychosis (H2), and self-stigma predicts decreased social functioning (H3), which in turn predicts greater severity of psychosis (H4). Moreover, in the presence of these mediators (self-stigma and social functioning), the direct effect of psychological inflexibility on the severity of psychosis (H5) would disappear. 

Following the testing of this mediation model, the second objective of this study was to establish which facets (dimensions) of internalized stigma are mediating in their indirect effects; that is, which dimensions predict deterioration in social functioning, leading to greater severity of psychosis. 

## 2. Materials and Methods

### 2.1. Participants

The study included 103 adult outpatients (M = 49.68, SD = 12.28 years) with chronic psychosis from the Mental Health Public Service in Asturias, Spain. The patients consisted of 60 males and 43 females. Participants diagnosed according to ICD-10 criteria [37], as determined by the ICD-10 codes F20–F29, were recruited (Table 1). Subjects were excluded from the analysis if they presented with a sign of severe physical illness, an episode of acute psychosis, an intellectual disability, or a primary diagnosis of substance dependence. 

Written informed consent was obtained from all subjects after the procedures had been fully explained. This study was performed according to the Declaration of Helsinki and was approved by the Research Ethical Committee of the Principality of Asturias, Spain. 

### 2.2. Measures

The Acceptance and Action Questionnaire–II (AAQ-II) [38] was used to assess psychological inflexibility. It consists of 7 items in a 7-point Likert scale. The items reflect an unwillingness to experience unwanted emotions and thoughts, and interference of these internal events with daily functioning (e.g., “I’m afraid of my feelings”). Higher scores indicate greater psychological inflexibility and less persistence with life goals. The AAQ-II has demonstrated good internal consistency (α = 0.88) [39].

Internalized stigma was measured using the Internalized Stigma of Mental Illness Scale (ISMI) [40], which has been widely used among sample patients with psychosis [41]. ISMI consists of five dimensions of mental illness self-stigma, including alienation, stereotype endorsement, discrimination experience, social withdrawal, and stigma resistance (reverse-scored). ISMI includes self-statements regarding one’s beliefs about having a mental illness, using 29-item Likert-type items on a 4-point scale (1–4), where higher scores indicate greater internalized stigma (e.g., “people ignore me or take me less seriously just because I have a mental illness”). Scores from the five dimensions are totaled to create a composite ISMI-29 score, with values ranging from 29 to 116. Higher scores indicate greater levels of mental illness self-stigma. Previous research [40] has shown that ISMI-29 exhibited good internal consistency (α = 0.87) and strong construct validity. ISMI was used in its Spanish version [42].

The Social Functioning Scale (SFS) [43], specifically designed for use with patients with psychosis, was used to evaluate the seven areas of functioning: withdrawal (i.e., social avoidance); interpersonal behavior (e.g., number of friends); pro-social activities (e.g., engagement in social activities); recreation (e.g., engagement in pastimes); independence–performance; independence–competence; and employment (i.e., engagement in productive employment). The total SFS score is the sum of the seven domain scores, with a range of 0–226. A higher score indicates a higher level of social functioning. SFS has been shown to be valid, reliable, and sensitive to change. Cronbach’s alpha is between 0.69 and 0.80 in every subscale.

The Clinical Global Impression–Schizophrenia Scale (CGI-SCH) [44] was used to evaluate the overall severity of schizophrenia. The CGI scale comprises 2 categories: severity of illness and degree of change. The severity of illness category evaluates the situation during the week prior to the assessment, while the degree of change category evaluates the change from the previous evaluation. Each category contains five different ratings (positive, negative, depressive, cognitive, and global) that are evaluated using a seven-point ordinal scale. The CGI-SCH global score assesses the global severity of the disorder, including symptoms and interference with functioning. In this study, only the severity of illness was assessed. CGI-SCH is a valid and reliable instrument for assessing illness severity and treatment response in psychosis. The inter-rater reliability for global scores was 0.79. 

### 2.3. Procedure

Each responsible psychiatrist evaluated the participants using CGI-SCH. The other instruments (ISMI, AAQ-II, and SCS) were administered by a nurse who specialized in mental health, with more than 11 years of clinical experience.

The nurse ensured that all participants understood the questions included in the administered measures, and when necessary, that assistance was provided to facilitate their understanding.

### 2.4. Statistical Methods

To examine the hypothesized mediation model, we used the PROCESS macro for SPSS (IBM Corporation, Armonk, NY, USA) [45], Model 6, which postulates a mediation model with two sequential mediators. The indirect effect (IV) of psychological inflexibility on the severity of psychosis (DV) was evaluated via the internalized stigma and social functioning mediators. The indirect effect was calculated using 10,000 bootstrap samples for the bootstrap confidence intervals (CIs), corrected for skewness. An indirect effect is considered statistically significant if the CI set (CI at 95%) does not include 0. If 0 is included in the CI, the null hypothesis states that the indirect effect is equal to 0; that is, there is no association between the variables involved [45].

The PROCESS macro for SPSS was also used to study the facets of internalized stigma that predict a decrease in social functioning, leading to greater severity of psychosis. As in the previous objective, the indirect effect was calculated using 10,000 bootstrap samples for the bootstrap confidence intervals (ICs). In addition to IV (psychological inflexibility) and DV (severity of psychosis), the dimensions of internalized stigma were included as five parallel mediators: alienation (M1); stereotype approval (M2); discriminatory experience (M3); social withdrawal (M4); resistance to stigma (M5); and social functioning (M6) as a sequential mediator. 

## 3. Results

### 3.1. Descriptive Analysis

The means, standard deviations, and correlations of the study’s dependent, independent, and mediator variables are shown in Table 2.

### 3.2. Hypothesis Evaluation

First, the influence of PI on severity of psychosis was analyzed. With regard to Hypothesis 1, the results of the total effect of PI (IV) on the severity of psychosis (DV) confirmed that higher PI predicted greater severity (B = 0.37, *p* = 0.0001).

Hypothesis 2 was also confirmed, because (as expected) PI predicted self-stigma of individuals with chronic psychosis (B = 0.66, *p* < 0.00001). Furthermore, as posed in Hypothesis 3, this level of internalized stigma negatively predicted social functioning (B = −0.24, *p* = 0.039). Hypothesis 4 was met, because decreased social functioning predicted greater severity of psychosis (B = −0.28, *p* = 0.007). Likewise, higher PI predicted worse social functioning (B = −0.31, *p* = 0.009). Self-stigma did not predict the severity of psychosis (B = 0.23, *p* = 0.062).

With regard to Hypothesis 5, the results showed that the direct effect of PI on the severity of psychosis disappeared (B = 0.09, *p* = 0.460) when mediation of internalized stigma and social functioning was included in that relationship, as expected in our proposal. Model mediation, shown in Figure 2, had a completely standardized indirect effect of B = 0.279, BootSE = 0.103; Boot 95% CI = [0.089, 0.492]. 

With respect to the study’s second objective—analyzing the mechanisms by which internalized stigma influences social functioning—a mediation model similar to the one in Figure 2 was computed, but with the five internalized stigma dimensions as simultaneous mediators (not sequential mediators): alienation, stereotype endorsement, discriminatory experience, social withdrawal, and stigma resistance (Figure 3).

The results showed that only the stereotype endorsement and stigma resistance dimensions of ISMI were significant for predicting social functioning. Thus, although PI predicted all the dimensions of self-stigma (*p* < 0.0001), only stereotype endorsement and stigma resistance predicted social functioning. Higher stereotype endorsement predicted worse social functioning (B = −0.33, *p* = 0.004), and greater stigma resistance predicted better social functioning (B = 0.22, *p* = 0.021). Again, social functioning negatively predicted the severity of psychosis (B = −0.300, *p* = 0.007). In this mediation model, the direct effect of PI on the severity of psychosis disappeared (B = 0.08, *p* = 0.515).

## 4. Discussion

One of the objectives of this study was to examine the relationships between PI, self-stigma, social functioning, and the severity of psychosis in individuals with chronic psychosis. Furthermore, as the literature on self-stigma and its association with psychological inflexibility is still in its infancy, we explored whether PI was able to predict an association between self-stigma and the severity of psychotic symptoms. 

Regarding the evaluation of self-stigma, the study revealed abundant evidence of the salience of stigma in the lives of people with chronic psychosis. Self-stigma was related to social functioning and symptom severity in the persons evaluated. Psychological inflexibility also predicted self-stigma.

Before this study, other studies had already shown that PI is related to self-stigma in individuals diagnosed with schizophrenia [29]. However, to our knowledge, there were no empirical studies that explored the reasons for this relationship. As far as we know, this study is the first to show a finding, based on clinical samples, that the link between PI and the severity of psychotic symptoms is not direct, but is better explained by mediation of self-stigma and social functioning of those diagnosed. That is, the predictive power of psychological inflexibility on the severity of psychosis disappeared when we included self-stigma and social functioning as sequential mediators. It should also be mentioned that psychological inflexibility predicts decreased social functioning without the need to take self-stigma into account. Furthermore, self-stigma alone does not predict the severity of psychosis; this relationship must be mediated by social functioning.

In addition to confirming the negative influence of PI on the severity of psychosis, this study considered how psychological inflexibility could cause such repercussions. The repercussions include the harmful impacts that the processes underlying PI (e.g., cognitive fusion, avoidance, attachment to the conceptualized self, etc.) have on thoughts, feelings, emotions, and other internal events causing distress. This study demonstrated that people with an inflexible psychological style avoid and struggle to rid themselves of their negative self-evaluations (i.e., the content of self-stigma), and that this struggle diminishes even more their social functioning and negatively impacts on evaluations of disorder severity. Acceptance and Commitment Therapy could be an appropriate treatment for testing the value of this line of reasoning.

Like PI, stigma and self-stigma are also inherently rigid processes [46,47]. Cognitive fusion, one of the processes underpinning the PI model, occurs when one is strongly influenced by his/her own thoughts and thinks they accurately represent who he/she is as a person. This relationship with one’s own thoughts as if they were literal translations of reality reinforces behaviors directed at avoiding associated discomfort (e.g., isolation, avoiding situations in which one feels “strange”, rejecting healthy behaviors, etc.). Paradoxically, efforts to suppress unwanted thoughts often increase their frequency and intensity [48]. Thus, a vicious circle is established, which ends up invading important areas of the lives of those who experience psychosis. On this basis, a reduction in unwanted thoughts and feelings should not be the therapeutic goal. Rather, rehabilitation should be directed at reducing ineffective behaviors (not thoughts) and breaking with the tendency to consider that such thoughts and feelings are the reasons for the behaviors.

Promotion of psychological flexibility could be useful in buffering such relationships. People with chronic psychosis who show more psychological flexibility could learn how to relate differently to so-called self-stigma, because they would be more likely to experience self-stigmatizing thoughts (e.g., shame, guilt, etc.) as passing events in their minds [49], and not as thoughts that define who they are. Comparing the effects of ACT and psychoeducation on self-stigma, Fung et al. [15] observed that ACT had the most robust response in combatting the effects of self-stigma, while a program of psychoeducation had no impact on internalized stigma. ACT expands traditional cognitive behavioral therapy with a focus on helping patients to relate to their thoughts differently, while de-emphasizing the need to change or “restructure” specific contents [50]. In other words, the focus is more on the process of thinking, rather than on the content of thinking.

Aside from self-stigma, people with chronic psychosis who have more psychological flexibility would improve their social functioning. Psychological flexibility includes changing behavior that compromises personal or social functioning. As previous studies have shown [31], the inflexibility observed in patients with poor social functioning suggests a struggle with symptoms, which impedes patients from becoming the managers of their own lives and, in the long run, invades and disrupts all the facets that generally define their lives (e.g., work, intimate relations, family relations, etc.). Maintaining a balance between important life domains and being aware, open, and committed to behaviors that are congruent with deeply rooted values would not only help to improve social functioning, but also provide an integral approach to psychosis.

The results reported here lead us to doubt the feasibility of directly changing the form and frequency of self-devaluating or stigmatizing thoughts. Focusing on the underlying verbal processes of categorization, association, and evaluation, rather than on the specific topographical content of self-stigmatizing thoughts, could weaken the penetrating and rigid nature of stigma/self-stigma. As Mittal et al. [51] suggested in their review, interventions against stigma that concentrate on psychoeducation and reducing negative self-valuation should be broadened to include strategies that favor psychological flexibility, such as training in cognitive defusion techniques or mindfulness. There is good evidence of the feasibility and acceptability of ACT for people with psychosis [25,52].

In this study, the facets of stigma that exerted the mediating role between PI and the severity of psychosis were also analyzed. Our results showed that only the stereotype endorsement (r = −0.333) and stigma resistance (r = 0.224) factors were able to mediate in this relationship. Psychological inflexibility showed this influence by increasing the stereotype endorsement dimension (comprised of thoughts such as “I can’t contribute anything to society because I have a mental illness” or “The stereotypes of mental illness are applicable to me”) and, at the same time, diminishing the stigma resistance subscale (i.e., the capacity for resilience with respect to social stigma). Altogether, PI showed a capacity to strongly erode the social functioning of the participants.

Stereotype endorsement refers to the set of thoughts plagued with self-criticism and contempt for one’s capacity to feel like a useful member of the community. As in other studies, the results seem to confirm that self-stigma in psychosis is characterized by dissatisfaction with social relationships, high levels of stereotyping, withdrawal, and alienation [53]. This is a result that underlines both the need to change the credibility and importance of the self-critical thoughts that define self-stigma, and the need to promote self-compassion as a more adaptive way of relating with oneself and others. This type of strategy, characteristic of ACT, improves the psychological functioning of the participants [54] and could “remove” stigma and exclusion from people who live with psychosis. 

In brief, the development of strategies related to cognitive defusion—learning to relate to stigmatizing thoughts as just thoughts (not as literally true events—and self-compassionate acceptance—being open to non-defensive experience—would help to reduce the domain and useless functions of self-stigma. The same strategies would also help to improve the social functioning of the participants. Through these processes, people can become involved in more important, more valuable behaviors that were previously avoided due to due to the rigidity that also explains the pernicious effects of self-stigma in mental health.

Some limitations in our study should be considered. Due to the study’s cross-sectional nature, causality among the variables could not be established, and the interpretation of the proposed paths should be regarded with caution. A longitudinal study of the model is required to confirm the observed links between psychological inflexibility, self-stigma, and social functioning. Using intervention-based designs, future studies could confirm the mediating mechanisms by examining whether the fostering of psychological flexibility among people with mental illness deactivates the damaging impact of self-stigma, improves social functioning, and contributes to the recovery of people with chronic psychosis. 

In spite of these limitations, the present study provides support for the potential of psychological flexibility interventions, such as ACT, in promoting personal recovery. The study also extends the existing literature by suggesting that the processes of psychological inflexibility—especially cognitive fusion—make it possible to explain the reasons for the harmful impact of self-stigma on people with chronic psychosis. 

## 5. Conclusions

Psychological inflexibility was presented in this study as a potential process for explaining the negative impact of self-stigma in psychosis. This study also demonstrates that the effect of this process on the severity of psychosis occurs through its influence on social functioning. According to our results, consequences such as isolation, invalidity, and decreased social adjustment result from self-stigma induced by an inflexible psychological style. The use of therapies that promote social connection by changing the patient’s focus could be effective. ACT would help to normalize the experience of disturbing thoughts and any other psychotic symptoms; as a result, active participation in areas of life that are genuinely important and valuable should be encouraged.

## Figures and Tables

**Figure 1 ijerph-18-12376-f001:**
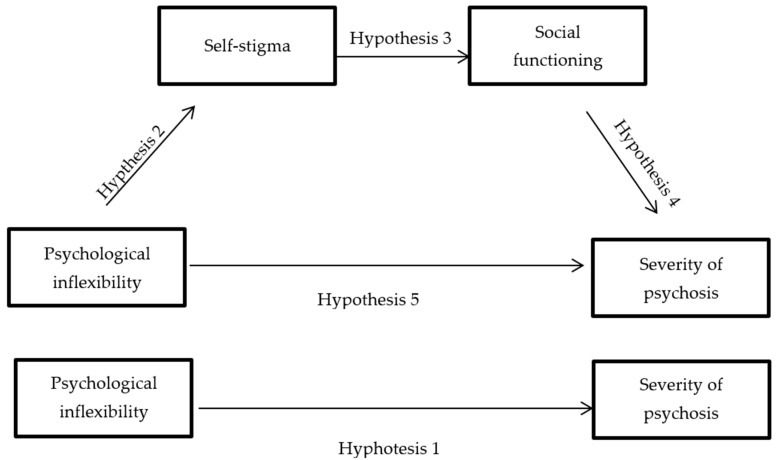
Hypothesized mediation model.

**Figure 2 ijerph-18-12376-f002:**
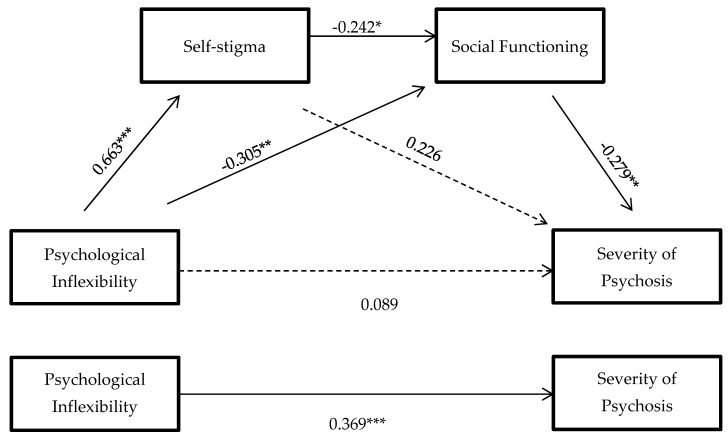
Hypothesized mediation model: indirect effect of psychological inflexibility on severity of psychosis through internalized stigma and social functioning, and total effect (standardized regression coefficients): * *p* < 0.05, ** *p* < 0.01, *** *p* ≤ 0.0001.

**Figure 3 ijerph-18-12376-f003:**
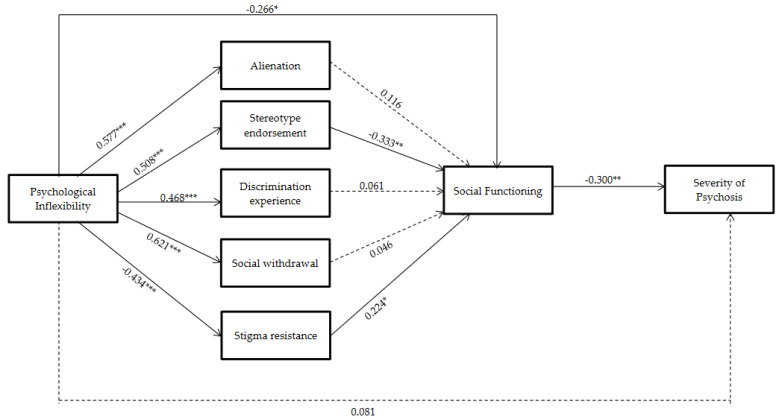
Hypothesized mediation model: indirect effect of psychological inflexibility on the severity of psychosis through social functioning and internalized stigma factors (standardized regression coefficients): * *p* < 0.05, ** *p <* 0.01, *** *p* < 0.0001.

**Table 1 ijerph-18-12376-t001:** Main participant characteristics (N = 103).

		n or M	% or SD
ICD-10 diagnosis	Paranoid schizophrenia (F20.0)	65	63.1
	Undifferentiated schizophrenia (F20.3)	2	1.9
	Residual schizophrenia (F20.5)	3	2.9
	Simple-type schizophrenia (F20.6)	4	3.9
	Schizophrenia, unspecified (F20.9)	2	1.9
	Schizotypal disorder (F21)	6	5.8
	Delusional disorder (F22)	13	12.6
	Acute and transient psychotic disorders (F23)	2	1.9
	Schizoaffective disorder (F25)	5	4.9
	Other schizophrenia (F28)	1	1
Marital status	Single	65	63.1
	Married/with partner	19	18.5
	Separated/divorced	17	16.5
	Widow	2	1.9
Living arrangements	Alone	28	27.2
	Family of origin	58	56.3
	Shared flat	2	1.9
	With partner	12	11.7
	Residence/supervised flat	3	2.9
Education	No education	7	6.8
	Primary	32	31.1
	Secondary	50	48.5
	University	14	13.6
Employment situation	Housework	5	4.9
	Inactive (unemployed)	20	19.4
	Active (employed)	17	16.5
	Inactive (employment disability)	43	41.7
	Inactive (retired)	6	5.8
	Other	12	11.7
Substance use	No	49	47.6
	Yes	40	38.8
	Ex-user	14	13.6
Age at onset of psychosis (years)		30.64	11.75
Diagnosis (years)		19.04	10.88
Pharmacological treatment (years)		17	10.43
Number of hospitalizations		2.21	3.83

**Table 2 ijerph-18-12376-t002:** Descriptive statistics and bivariate correlations for study variables.

	Mean	SD	1	2	3	4	5	6	7	8
1. Psychological inflexibility	26.12	11.23								
2. Self-stigma	1.98	0.61	0.66 **							
3. Alienation	2.06	0.89	0.58 **	0.89 **						
4. Stereotype endorsement	1.74	0.59	0.51 **	0.75 **	0.59 **					
5. Discrimination experience	2.02	0.86	0.47 **	0.83 **	0.68 **	0.56 **				
6. Social withdrawal	1.89	0.84	0.62 **	0.85 **	0.76 **	0.50 **	0.63 **			
7. Stigma resistance	2.18	0.67	−0.43 **	−0.60 **	−0.42 **	−0.40 **	−0.30 **	−0.37 **		
8. Social functioning	98.67	10.96	−0.47 **	−0.44 **	−0.32 **	−0.48 **	−0.27 **	−0.33 **	0.42 **	
9. Severity of psychosis	4.18	1.06	0.37 **	0.41 **	0.37 **	0.38 **	0.31 **	0.37 **	−0.19 *	−0.42 **

Note. *n* = 103; ** *p ≤* 0.001, except * *p* = 0.059.

## Data Availability

The data presented in this study are available on request from the corresponding author. The data are not publicly available due to privacy and ethical reasons.

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
