# Peer review of "Psychological Inflexibility in People with Chronic Psychosis: The Mediating Role of Self-Stigma and Social Functioning"

_ijerph, 2021, doi:10.3390/ijerph182312376_

Round 1
Reviewer 1 Report
This is an important and well manuscript. Research methodology is scientifically sound, and results pose important implications for treatment of psychotic patients.
Nevertheless, a few changes would improve the overall quality of the manuscript:
- Please provide clarification on the definition of “people with psychosis”.
- Stigma and Self-stigma definition need better articulation and clarification.
- Please include a list of diagnoses contained in the ICD-10 codes F20-F29.
- Table 1. Columns do not match some variables. Numeric variables should read Mean and SD. Also, replace F with n.
- Please provide examples of items for each measure.
- Please provide internal consistency scores for the SFS and CGI-SCH scales.
- Procedures need to be further detailed.
- Table 2. Is DT standard deviation? Please include notes for significance: *<0.05; **<0.001
- Please include a limitations section.
- Please include an implication section, discussing results in terms of public mental health measures and intervention policies.
Best wishes.
Author Response
We want to thank you for your time and suggestions. We have revised our manuscript following all your interesting comments.
We have clarified the expression "people with psychosis", expanded the notion of stigma, and also included the list of diagnoses presented by the study sample. We have modified and corrected tables 1 and 2 and we have also included a section on limitations and implications.
All these changes have been marked in red in our manuscript.
We look forward to responding to your comments and again, we want to sincerely thank you for your interest in our manuscript.

Reviewer 2 Report
After reading the article, I consider it to be an original work, which adequately reviews the state of the art in the literature. The design of the study is cross-sectional, which means that the conclusions reached must be considered with caution.
Perhaps this is an aspect that the authors should highlight as a possible limitation of the results in the final discussion section.
Despite this, I consider the results to be valuable and open the door to detecting significant variables when introducing psychological interventions based on ACT, which are potentially useful in this field.
1. This paper takes an original approach to the mediating variables between psychological inflexibility and severity of psychosis. The authors argue that psychological inflexibility predicts symptom severity based on the mediation of self-stigma, which in turn is related to some dimensions of social functioning.
2. The topic raised is relevant to the field of study of the relationship between psychological inflexibility and self-stigma in chronic psychosis. Until now, self-stigma has been related to the severity of symptomatology and social functioning in people with chronic psychosis. However, as the authors point out, psychological inflexibility is also related to and predictive of self-stigma, but until the present research, it was not known why. From this perspective, this is the first empirical work to address the variables that may mediate this relationship in the field of psychosis.
3. In relation to other published materials, an original concrete result is presented on the mediating role of the variables self-stigma and social functioning (more specifically, high scores on stereotiped endorsement predict worse social functioning and higher stigma resistance predicts better social functioning). Specifically, these two dimensions of self-stigma predict poorer social functioning, which in turn predicts chronicity of symptoms in people with chronic psychosis.
Based on these results, the authors propose therapeutic interventions from an ACT perspective in which people with more psychological flexibility could learn to relate differently to the stereotype endorsement and stigma resistance dimensions of self-stigma, considering these self-stigmatising thoughts in a more transient way, and not as absolute thoughts. In this way, the aim would not be to eliminate these thoughts, but to learn to live with them; the authors suggest that through defusion or mindfulness techniques, and acceptance and self-compassion, these people could diminish the paralysing function of self-stigma, and also improve their level of social functioning. I believe this is an original therapeutic approach, an alternative to the more traditional second-generation cognitive-behavioural interventions.
4.I believe that the methodology used is adequate for a study of these characteristics. As I pointed out in my previous assessment, the only suggestion I would make is to point out in the limitations section that this is a cross-sectional study, and that the results relating to mediating variables would carry much more weight if they were carried out in a study with pre-post scores. In other words, they could consider future research in which they would intervene with this group using the techniques they point out (defusion, mindfulness, self-compassion) to analyse the possible changes seen in symptomatology and thus ratify the role of the mediating variables proposed.
5. I consider that the conclusions are consistent with the results obtained, presenting arguments related to the applicability of these results to the field of psychological intervention from the perspective of psychological inflexibility and ACT.
6. I consider the references to be adequate and sufficiently up to date.
7. In Figure 2 the fact that the standardised regression coefficient score .226 overlaps with the dashed arrow should be improved.
Author Response
Thank you very much for your interest in our manuscript, for your time and your kind comments.
We have corrected the error observed in figure 2 and, following your suggestion, a section of limitations. These changes, along with others suggested by other reviewers, are detailed in the manuscript in red.
Again, thank you very much for your words and interest.
